# Lifelong Reinforcement Learning with Modulating Masks

**Eseoghene Ben-Iwhiwhu**                                   *e.ben-iwhiwhu@lboro.ac.uk*
*Department of Computer Science, Loughborough University, UK*

**Saptarshi Nath**                                          *s.nath@lboro.ac.uk*
*Department of Computer Science, Loughborough University, UK*

**Praveen K. Pilly**                                        *pkpilly@hrl.com*
*HRL Laboratories, LLC, Malibu, CA, 90265, USA*

**Soheil Kolouri**                                          *soheil.kolouri@vanderbilt.edu*
*Department of Computer Science, Vanderbilt University, Nashville, TN, USA*

**Andrea Soltoggio**                                        *a.soltoggio@lboro.ac.uk*
*Department of Computer Science, Loughborough University, UK*

**Reviewed on OpenReview:** *https://openreview.net/forum?id=V7tahqGrOq*

## Abstract

Lifelong learning aims to create AI systems that continuously and incrementally learn during a lifetime, similar to biological learning. Attempts so far have met problems, including catastrophic forgetting, interference among tasks, and the inability to exploit previous knowledge. While considerable research has focused on learning multiple supervised classification tasks that involve changes in the input distribution, lifelong reinforcement learning (LRL) must deal with variations in the state and transition distributions, and in the reward functions. Modulating masks with a fixed backbone network, recently developed for classification, are particularly suitable to deal with such a large spectrum of task variations. In this paper, we adapted modulating masks to work with deep LRL, specifically PPO and IMPALA agents. The comparison with LRL baselines in both discrete and continuous RL tasks shows superior performance. We further investigated the use of a linear combination of previously learned masks to exploit previous knowledge when learning new tasks: not only is learning faster, the algorithm solves tasks that we could not otherwise solve from scratch due to extremely sparse rewards. The results suggest that RL with modulating masks is a promising approach to lifelong learning, to the composition of knowledge to learn increasingly complex tasks, and to knowledge reuse for efficient and faster learning.

## 1 Introduction

Lifelong reinforcement learning (LRL) is a recent and active area of research that seeks to enhance current RL algorithms with the following capabilities: learning multiple tasks sequentially without forgetting previous tasks, exploiting previous tasks knowledge to accelerate learning of new tasks, or building solutions to increasingly complex tasks. The remarkable progress of RL in recent years, stemming in particular from the introduction of deep RL (Mnih et al., 2013), has demonstrated the potential of algorithms that can learn to act in an environment with complex inputs and rewards. However, the limitation of learning only one task means that such algorithms have a narrow focus, and may not scale to complex tasks that first require learning of sub-tasks. It is arguable that acquiring knowledge of multiple tasks over a lifetime, generalizing across such tasks, exploiting acquired knowledge to master new tasks more quickly, or composing simple

tasks to solve more complex tasks are necessary to develop more powerful AI systems that better mimic biological intelligence.

LRL shares some of the objectives of lifelong supervised learning (LSL) (Parisi et al., 2019; Hadsell et al., 2020; De Lange et al., 2021), but due to the unique objective and domains of RL, it has developed into a separate field (Khetarpal et al., 2020). For example, tasks in LRL can vary across reward function, input distribution or transition function, while tasks in LSL largely vary in the input distribution. A variety of recently developed approaches in LSL can be placed under three categories: memory methods, synaptic consolidation methods, and parameter isolation or modular methods. In LRL, Khetarpal et al. (2020) propose a taxonomy in which approaches can be classified as explicit knowledge retention, leveraging shared structures, and learning to learn. It can be appreciated that LRL exploits most advances in LSL while also developing RL-specific algorithms.

Among the LSL categories mentioned above, parameter isolation methods have shown state-of-the-art results in classification with approaches such as Progressive Networks (Rusu et al., 2016; Yoon et al., 2018), PackNet (Mallya & Lazebnik, 2018), Piggyback (Mallya et al., 2018) and Supermasks (Wortsman et al., 2020). The key concept in such methodologies is to find smart and efficient ways to isolate parameters for each task. In Mallya & Lazebnik (2018), redundancies in large networks are exploited to generate masks via iterative pruning, and thus free parameters to be used only for specific tasks, thus "packing" multiple tasks in one single network. Other approaches instead use the concept of directly learned masks (Zhou et al., 2019; Ramanujan et al., 2020; Wortsman et al., 2020) to select parts of a backbone network for each task. Instead of learning the parameters of a backbone network, which are set randomly, masking approaches focus on learning a multiplicative (or modulating) mask for each task. Masks can be effective in binary formats, enabling or disabling parts of the backbone network while requiring low memory. Interestingly, masks can be interpreted as *modulatory* mechanisms with biological inspiration (Kudithipudi et al., 2022) because they exploit a gating mechanism on parameters or activations.

Surprisingly, while the application of masks has been tested extensively in LSL for classification, very little is known on their effectiveness in LRL. Also, the current mask methods lack the ability of exploit knowledge from previously learned task (forward transfer), since each mask is derived independently. This study introduces the use of directly learned masks with policy optimization algorithms, specifically PPO (Schulman et al., 2017) and IMPALA (Espeholt et al., 2018). We explore (1) the effectiveness of masks to maximize lifelong learning evaluation metrics, and (2) the suitability of masks to reuse knowledge and accelerate learning (forward transfer). To demonstrate the first point, we test the approach on RL curricula with the Minigrid environment (Chevalier-Boisvert et al., 2018), the CT-graph (Soltoggio et al., 2019; 2023), Metaworld (Yu et al., 2020), and ProcGen (Cobbe et al., 2020), and assess the lifelong learning metrics (New et al., 2022; Baker et al., 2023) when learning multiple tasks in sequence. To demonstrate the second point, we exploit linear combinations of masks and investigate learning with curricula in which tasks have similarities and measure the forward transfer. The investigation of the second point gives rise to the understanding of the effectiveness of mask knowledge reuse in RL, the initial parameter configuration ideal for mask reuse, and the benefits obtained when task curriculum grow in complexity or contains interfering tasks. The proposed method assumes a task oracle, but the assumption can be eliminated through the introduction of task detection methods into the system as discussed in Section 7. To ensure reproducibility, the hyper-parameters for the experiments are reported in Appendix B. The code is published at `https://github.com/dlpbc/mask-lrl`.

**Contributions.** To the best of our knowledge, this is the first study to (1) introduce learned modulating masks in LRL and show competitive advantage with respect to well established LRL methods (2) show the exploitation and the composition of previously learned knowledge in a LRL setup with learned modulating masks. In addition to a baseline masking approach with independent mask search, we introduced two new incremental mask composition approaches (LC and BLC) with different initialization of linear combination parameters. The analysis reveals different properties in the exploitation of previous knowledge while learning new tasks, either favoring knowledge re-use or exploration.

## 2 Related work

### 2.1 Deep Reinforcement Learning (DeepRL)

The use of deep networks as function approximators in reinforcement learning (RL) has garnered widespread adoption, showcasing results such as learning to play video games (Mnih et al., 2013; Van Hasselt et al., 2016; Shao et al., 2018; Kempka et al., 2016; Mnih et al., 2016; Babaeizadeh et al., 2017; Espeholt et al., 2018), and learning to control actual and simulated robots (Lillicrap et al., 2016; Schulman et al., 2015; 2017; Fujimoto et al., 2018; Haarnoja et al., 2018). These algorithms enable the agent to learn how to solve a single task in a given evaluation domain. In a lifelong learning scenario with multiple tasks, they suffer from lifelong learning challenges such as catastrophic forgetting. Nevertheless, they serve as the basis for lifelong learning algorithms. For example, DQN (Mnih et al., 2013) was combined with the elastic weight consolidation (EWC) algorithm to produce a lifelong learning DQN agent (Kirkpatrick et al., 2017; Kessler et al., 2022).

### 2.2 Lifelong (Continual) Learning

Several advances have recently been introduced in lifelong (continual) learning (Van de Ven & Tolias, 2019; Parisi et al., 2019; Hadsell et al., 2020; De Lange et al., 2021; Khetarpal et al., 2020), addressing challenges such as maintaining performance on previously learned tasks while learning a new task (overcoming forgetting), reusing past knowledge to rapidly learn new tasks (forward transfer), improving performance on previously learned tasks from newly acquired knowledge (backward transfer), the efficient use of model capacity to reduce intransigence, and reducing or avoiding interference across tasks (Kessler et al., 2022). A large body of work focused on lifelong learning in the supervised learning domain and overcoming the challenge of forgetting (Mendez et al., 2022).

Lifelong learning algorithms can be clustered into key categories such as synaptic consolidation approaches (Kirkpatrick et al., 2017; Zenke et al., 2017; Aljundi et al., 2018; Kolouri et al., 2019), memory approaches (Lopez-Paz & Ranzato, 2017; Zeng et al., 2019; Chaudhry et al., 2019; Rolnick et al., 2019; Lin et al., 2022), modular approaches (Rusu et al., 2016; Mallya & Lazebnik, 2018; Mallya et al., 2018; Wortsman et al., 2020; Mendez et al., 2022) or a combination of the above. Synaptic consolidation approaches tackle lifelong learning by discouraging the update of parameters useful for solving previously learned tasks through a regularization penalty. Memory approaches either store and replay samples of previous and current tasks (from a buffer or a generative model) during training (Lopez-Paz & Ranzato, 2017; Chaudhry et al., 2019; Guo et al., 2020; von Oswald et al., 2020) or project the gradients of the current task being learned in an orthogonal direction to the gradients of previous tasks (Zeng et al., 2019; Farajtabar et al., 2020; Saha et al., 2021; Lin et al., 2022). Memory methods aim to keep the input-output mapping for previous tasks unchanged while learning a new task. Modular approaches either expand the network as new tasks are learned (Rusu et al., 2016; Yoon et al., 2018; Martin & Pilly, 2019) or select sub-regions of a fixed-sized network (via masking) for each tasks (Wortsman et al., 2020). The masks can be applied to (i) the neural representations (Sokar et al., 2021; Serra et al., 2018), (ii) or to synapses, where they are directly learned (Wortsman et al., 2020) or derived through iterative pruning (Mallya & Lazebnik, 2018). The masks can be viewed as a form induced sparsity in the neural lifelong learner (Von Oswald et al., 2021; Hoefler et al., 2021).

In LRL, the learner is usually developed by combining a standard deep RL algorithm, either on-policy or off-policy (for example, DQN, PPO (Schulman et al., 2017), or SAC (Haarnoja et al., 2018)), with a lifelong learning algorithm. CLEAR (Rolnick et al., 2019) is a lifelong RL that combines IMPALA with a replay method and behavioral cloning. Progress & Compress (Schwarz et al., 2018) demonstrated the combination of IMPALA with EWC and policy distillation techniques. Other notable methods include the combination of a standard deep RL agent (SAC) with mask derived from pruning in PackNet (Mallya & Lazebnik, 2018; Schwarz et al., 2021) as demonstrated in the Continual World robotics benchmark (Wołczyk et al., 2021). In addition, Mendez et al. (2022) developed an algorithm combining neural composition, offline RL, and PPO to facilitate knowledge reuse and rapid task learning via functional composition of knowledge. To tackle a lifelong learning scenario with interference among tasks, current methods (Kessler et al., 2020; 2022) employ a multi-head policy network (i.e., a network with shared feature extractor layers connected to different

output layers/heads per task) that combine a standard RL algorithm (e.g. DQN or SAC) with synaptic consolidation methods. In another class of LRL approach, Kaplanis et al. (2018) demonstrated a lifelong RL agent that combined DQN with multiple-time scale learning at the synaptic level (Benna & Fusi, 2016), and the model was adapted to multiple timescale learning at the policy level (Kaplanis et al., 2019) via the combination of knowledge distillation and PPO.

## 2.3 Modulation

Neuromodulatory processes (Avery & Krichmar, 2017; Bear et al., 2020) enable the dynamic alteration of neuronal behavior by affecting synapses or the neurons connected to synapses. Modulation in artificial neural networks (Fellous & Linster, 1998; Doya, 2002) draws inspiration from modulatory dynamics in biological brains that have proven particularly effective in reward-based environments (Schultz et al., 1997; Abbott & Regehr, 2004), in the evolution of reward-based learning (Soltoggio et al., 2008; 2018), and in meta RL (Ben-Iwhiwhu et al., 2022). Modulatory methods in lifelong learning are set up as masks that alter either the neural activations (Serra et al., 2018) or weights of neural networks (Mallya & Lazebnik, 2018; Mallya et al., 2018; Wortsman et al., 2020; Koster et al., 2022). The key insight is the use of a modulatory mask (containing binary or real values) to activate particular network sub-regions and deactivate others when solving a task. Therefore, each task is solved by different sub-regions of the network. For each task, PackNet (Mallya & Lazebnik, 2018) trains the model based on available network capacity and then prunes the network to select only parameters that are important to solving the task. A binary mask representing important and unimportant parameters is then generated and stored for that task, while ensuring that important parameters are never changed when learning future tasks. PiggyBack (Mallya et al., 2018) keeps a fixed untrained backbone network, while it trains and stores mask parameters per task. During learning or evaluation of a particular task, the mask parameters for the task are discretized and applied to the backbone network by modulating its weights. The Supermask (Wortsman et al., 2020) is a generalization of the PiggyBack method that uses a k-winner approach to create sparse masks.

## 3 Background

### 3.1 Problem Formulation

A reinforcement learning problem is formalized as a Markov decision process (MDP), with tuple $\mathcal{M} = \langle \mathcal{S}, \mathcal{A}, p, r, \gamma \rangle$, where $\mathcal{S}$ is a set of states, $\mathcal{A}$ is a set of actions, $p : \mathcal{S} \times \mathcal{A} \to \mathcal{S}$ is a transition probability distribution $p(s_{t+1}|s_t, a_t)$ of the next state given the current state and action taken at time $t$, $r : \mathcal{S} \times \mathcal{A} \to \mathbb{R}$ is a reward function that produces a scalar value based on the state and the action taken at the current time step, $\gamma \to [0, 1]$ is the discount factor that determines how importance future reward relative to the current time step $t$. An agent interacting in the MDP behaves based on a defined policy (either a stochastic $\pi$ or a deterministic $\mu$ policy). The objective of the agent is to maximize the total reward achieved, defined as an expectation over the cumulative discounted reward $\mathbb{E}[\sum_{t=0}^{\infty} \gamma^t r(s_t, a_t)]$. It achieves this by learning an optimal policy.

In a lifelong learning setup, a lifelong RL agent is exposed to a sequence of tasks $\mathcal{T}$. Given $N$ tasks, the agent is expected to maximize the RL objective for each task in $\mathcal{T} = \{\tau_1, \tau_2, \ldots, \tau_n\}$. As the agent learns one task after another in the sequence, it is required to maintain (avoid forgetting) or improve (backward transfer of knowledge) performance on previously learned tasks. A desired property of such an agent is the ability to reuse knowledge from previous tasks to rapidly learn the current or future tasks.

### 3.2 Modulating masks

The concept of modulating masks has recently emerged in the area of supervised classification learning (Mallya et al., 2018; Serra et al., 2018; Ramanujan et al., 2020; Wortsman et al., 2020; Sokar et al., 2021; Koster et al., 2022). Masks work by modulating the weights or the neural representations of a network, and they can be directly learned (optimized) or derived via iterative pruning (Hoefler et al., 2021), with a goal of implementing sparsity in the network. The learned mask approach is employed in this work, and masks are used to modulate the weights of a network. By modulating the weights, sub-regions of the network

is activated, while other regions are deactivated. Given a network with layers $1, \ldots L$, with each layer $l$ containing parameters $W^l \in \mathbb{R}^{m \times n}$, for each layer $l$, a score parameter $S^l \in \mathbb{R}^{m \times n}$ is defined.

During a forward pass (in training or evaluation), a binary mask $M^l \in \{0,1\}^{m \times n}$ is generated from $S^l$ based on an operation $g(S^l)$ according to one of the following: (i) a threshold $\epsilon$ (where values greater than $\epsilon$ are set to 1, otherwise 0) (Mallya et al., 2018), or (ii) top-k values (the top $k\%$ values in $S^l$ yields 1 in $M^l$, while the rest are set to 0) (Ramanujan et al., 2020), or (iii) probabilistically sampled from a Bernoulli distribution, where the $p$ parameter for the distribution is derived from the sigmoid function $\sigma(S^l)$ (Zhou et al., 2019). An alternative approach is the generation of ternary masks (i.e., with values $\{-1, 0, 1\}$) from $S^l$, as introduced in Koster et al. (2022). The binary or ternary mask modulates the layer's parameters (or weights), thus activating only a subset of the $W^l$.

Given an input sample $\mathbf{x} \in \mathbb{R}^m$, a forward pass through the layer is given as $f(\mathbf{x}, W^l, S^l) = (W^l \odot M^l) \cdot \mathbf{x}$, where $\odot$ is the element wise multiplication operation. Only the score parameters $S^l$ are updated during training, while the weights $W^l$ are kept fixed at their initial values. For brevity, the weights and scores across layers are denoted as $W$ and $S$.

The training algorithm updating $S^l$ depends on the operation used to generate the binary mask. When the binary masks are generated from Bernoulli sampling, standard backpropagation is employed. However, in the case of thresholding or selecting top $k\%$, an algorithm called edge-popup is employed, which combines backpropagation with the straight-through gradient estimator (i.e., where the gradient of the binary mask operation is set to identity to avoid zero-value gradients) (Bengio et al., 2013; Courbariaux et al., 2015).

In a lifelong learning scenario with multiple tasks, for each task $\tau_k$, a score parameter $S_k$ is learned and used to generate and store the mask $M_k$. During evaluation, when the task is encountered, the mask learned for the task is retrieved and used to modulate the backbone network. Since the weights of the network are kept fixed, it means there is no forgetting. However, this comes at the cost of storage (i.e., storing a mask for each task).

## 4 Methods

We first introduce the adaptation of modulating masks to RL algorithms (Section 3.2). Following, we hypothesize that previously learned masks can accelerate learning of unseen tasks by means of a linear combination of masks (Section 4.2). Finally, we suggest that continuous masks might be necessary to solve RL tasks in continuous environments (Section 4.3).

### 4.1 Modulatory masks in Lifelong RL

We posit that a stochastic control policy $\pi_{\theta, \Phi}$ can be parameterized by $\theta$, the weights of a fixed backbone network, and $\Phi = \{\phi_1 \ldots \phi_k\}$, a set of mask score parameters for tasks $1 \ldots k$. $\phi_k$ are the scores of all layers of the network for task $k$, comprising the layers $1 \ldots L$, i.e., $\phi_k = \{S_k^1 \ldots S_k^L\}$. The weights $\theta$ of the network are randomly initialized using the signed Kaiming constant method (Ramanujan et al., 2020), and are kept fixed during training across all tasks. The mask score parameters $\Phi$ are trainable, but only $\phi_k$ is trained when the agent is exposed to task $k$. To reduce memory requirements, Wortsman et al. (2020) discovered that masks can be reduced to binary without significant loss in performance. We test this assumption by binarizing masks using an element-wise thresholding function

$$g(\phi_k) = \begin{cases} 1 & \phi_{k,\{i,j\}} > \epsilon \\ 0 & \text{otherwise} \end{cases} \tag{1}$$

where $\epsilon$ is set to 0. The resulting LRL algorithm is described in Algorithm 1. To assess this algorithm, we paired it with the on-policy PPO (Schulman et al., 2017) algorithm and the off-policy IMPALA (Espeholt et al., 2018) algorithm.

---

**Algorithm 1** Lifelong RL Algorithm with modulating masks

---

**Require:** Number of tasks $N$, maximum training steps per task P
**Require:** Rollout length (steps) z
 1: Initialize policy $\pi_{\theta,\Phi}$
 2: **for** k $\leftarrow 1 \ldots$ N **do**
 3:     Set task $k$
 4:     Set $steps \leftarrow 0$
 5:     **while** step $<$ P **do**
 6:         Rollout experiences $\{(s_1, a_1, r_1, s_1') \ldots (s_z, a_z, r_z, s_z')\}$ using $\pi_{\theta,\phi_k}$.
 7:         Update $steps \leftarrow steps + z$
 8:         Compute loss $\mathcal{L}_k(\pi_{\theta,\phi_k})$ based on an RL algorithm objective.
 9:         Compute gradients $\nabla_{\phi_k}\mathcal{L}_k(\pi_{\theta,\phi_k})$ with respect to $\phi_k$.
10:         Update mask score parameter for task $k$, $\phi_k \leftarrow \phi_k - \alpha\nabla_{\phi_k}\mathcal{L}_k(\pi_{\theta,\phi_k})$
11:     **end while**
12:     Store mask score $\phi_k$ or the binary mask $g(\phi_k)$
13: **end for**

---

### 4.2 Exploiting previous knowledge to learn new tasks

One assumption in LRL, often measured with metrics such as forward and backward transfer (New et al., 2022), is that some tasks in the curriculum have similarities. Thus, previously acquired knowledge, stored in masks, may be exploited to learn new tasks. To test this hypothesis, we propose an incremental approach to using previously learned masks when facing a new task. Rather than learn a large number of masks and infer which mask is useful for solving a task at test time via gradient optimized linear combination, as in Wortsman et al. (2020), we instead start to exploit previous knowledge from any small number of masks combined linearly, plus a trainable random mask $\phi_{k+1} = \{S_{k+1}^1 \ldots S_{k+1}^L\}$. The intuition is that strong linear combination parameters can be discovered quickly if similarities are strong, otherwise more weight is placed on the newly trained mask.

A new mask at layer $l$ is given by

$$S^{l,lc} = \beta_{k+1}^l S_{k+1}^l + \sum_{i=1}^{k} \beta_i^l S_i^{l,*} \tag{2}$$

where $S_{k+1}^l \in \phi_{k+1}$ are the scores of layer $l$ in task $k+1$, $S^{l,lc}$ denotes the transformed scores for task $k+1$ after the linear combination (lc) operation, $S_i^{l,*}$ denotes the optimal scores for previously learned task $i$, and $\beta_1^l, \ldots \beta_{k+1}^l$ are the weights of the linear combination (at layer $l$). To maintain a normalized weighted sum, a Softmax function is applied to the linear combination parameters before they are applied in Equation 2. When no knowledge is present in the system, the first task is learned starting with a random mask. Task two is learned using $\bar{\beta}_1 = 0.5$, weighting task one's mask, and $\bar{\beta}_2 = 0.5$, weighting the new random mask. The third task will have $\bar{\beta}_1 = \bar{\beta}_2 = \bar{\beta}_3 = 0.33$, and so on. Note that $\bar{\beta}_k$ denotes a vector of size $L$ which contains the co-efficient parameters for task $k$ across all $L$ layers of the network (i.e., $\bar{\beta}_k = \{\beta_k^1 \ldots \beta_k^L\}$).

In short, two approaches can be devised. The first one in which each mask is learned independently of the others. Experimentation of this approach will determine the baseline capabilities of modulating masks in LRL. We name this *Mask Random Initialization* (Mask$_{\text{RI}}$). The second approach attempts to exploit the knowledge acquired so far during the curriculum learning by using a linear combination of masks to learn a new one. Experimentation of this second approach will determine the capabilities of modulating masks to exploit previously learned knowledge. We name this second approach *Mask Linear Combination* (Mask$_{\text{LC}}$). The idea is graphically summarized in Figure 1.

It can be noted that, as the number of known tasks increases, the relative weight of each mask decreases in Mask$_{\text{LC}}$. This could be a problem, particularly as the weight of the new random mask is reduced, possibly biasing the search excessively towards an average of previous policies. Therefore, we introduce a third approach that attempts to combine the benefits of both Mask$_{\text{RI}}$ and Mask$_{\text{LC}}$: we set the initial weight of

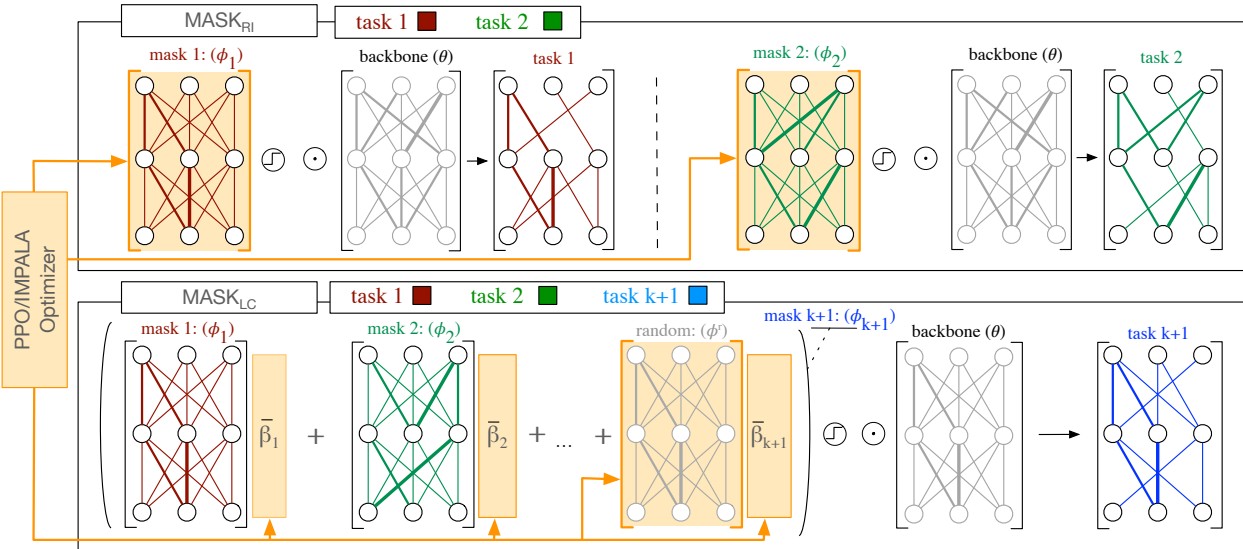

**Figure 1:** Graphical representations of the methods Mask$_{\text{RI}}$ (top) and Mask$_{\text{LC}}$ (bottom). Mask$_{\text{RI}}$ searches for a new mask for each new task independently. Mask$_{\text{LC}}$ attempts to exploit previous knowledge to learn a new task: known masks are linearly combined with a new randomly initialized mask while learning a new task. Gradient updates search for the parameters $\bar{\beta}_1, ..., \bar{\beta}_n$ and the new random mask.

the new random mask to 0.5, while the remaining 0.5 weight is shared by the masks of all known tasks. We name this third approach *Balanced Linear Combination* (Mask$_{\text{BLC}}$). It must be noted that the difference between Mask$_{\text{LC}}$ and Mask$_{\text{BLC}}$ is only in the initialization of weights when a new task is encountered, where $\bar{\beta}_{k+1} = 1/(k+1)$ in Mask$_{\text{LC}}$ and $\bar{\beta}_{k+1} = 0.5$ in Mask$_{\text{BLC}}$. However, the parameters $\bar{\beta}$ can be modified arbitrarily by backpropagation during training.

For both Mask$_{\text{LC}}$ and Mask$_{\text{BLC}}$, updates are made only to $S_{k+1}^l$ and $\bar{\beta}_i^l \ldots \bar{\beta}_{k+1}^l$ across each layer $l$. After the training on task $k+1$ is completed, the knowledge from the linear combination is consolidated into the scores for the current task $S_{k+1}^l$ by applying Equation 2. Therefore, the other masks and the linear combination parameters are no longer required. Algorithm 2 reports the forward pass operations for Mask$_{\text{LC}}$.

---

**Algorithm 2** Forward pass in network layer $l$ in Mask$_{\text{LC}}$

---

**Require:** Number of task learned so far $k$
**Require:** $S_1^* \ldots S_k^*$, $S_{k+1}$, $\bar{\beta}_1 \ldots \bar{\beta}_{k+1}$, $W$
1: **procedure** FORWARDPASS($\mathbf{x}$)
2:     **if** $k \leftarrow 0$ **then**
3:         Set $P \leftarrow S_1$
4:     **else**
5:         Compute the task score from linear combination: $P \leftarrow \bar{\beta}_{k+1} S_{k+1} + \sum_{i=1}^{k} \bar{\beta}_i S_i^*$
6:     **end if**
7:     Compute mask from score $M_{k+1} \leftarrow g(P)$
8:     Modulate weight $W^{mod} \leftarrow (W \odot M_{k+1})$
9:     Compute output $\mathbf{y} \leftarrow W^{mod} \cdot \mathbf{x}$
10:     **return y**
11: **end procedure**

---

### 4.3 Continuous Modulatory Masks for Continuous RL problems

In previous studies, binarization of masks was discovered to be effective in classification to reduce the memory requirement and improve scalability. As shown in our simulations, this approach was effective also in discrete

RL problems. However, in continuous action space RL environments, we discovered that binary masks did not lead to successful learning. It is possible that the quantization operation hurts the loss landscape in such environments since the input-output mapping is continuous to continuous values. Therefore, a modification of the supermask method can be devised to support the use of continuous value masks. In the thresholding mask generation operation (given a threshold $\epsilon$), the modified version becomes

$$g(\phi_k) = \begin{cases} \phi_{k,\{i,j\}} & \phi_{k,\{i,j\}} > \epsilon \\ 0 & \text{otherwise} \end{cases}.$$

(3)

Such a modification still maintains the ability to learn sparse masks, but replaces the unitary positive values with continuous values. The results of the empirical investigation of the binary and continuous masks in a continuous action space environment is reported in Section 5.4.

## 5 Experiments

The three novel approaches, $\text{Mask}_{\text{RI}}$, $\text{Mask}_{\text{LC}}$ and $\text{Mask}_{\text{BLC}}$, are tested on a set of LRL benchmarks across discrete and continuous action space environments. A complete set of hyper-parameters for each experiment is reported in Appendix B.

First, we employed the ProcGen benchmark (Cobbe et al., 2020) that consists of a set of video games with high diversity, fast computation, procedurally generated scenarios, visual recognition and motor control. The masking methods were implemented with IMPALA: $\text{Mask}_{\text{RI/LC/BLC}} + \text{IMPALA}$ (Espeholt et al., 2018) and tested following the ProcGen lifelong RL curriculum presented in Powers et al. (2022) (a subset of games in ProcGen). The properties of the benchmark make the curriculum challenging (e.g., high dimensional RGB observations and procedural generations of levels). The curriculum was designed to test generalization abilities: for each task, the lifelong evaluations are carried out using procedurally generated levels that are unseen by the agents during training. The masking methods were compared with baselines such as online EWC, P&C (Schwarz et al., 2018), CLEAR (Rolnick et al., 2019), and IMPALA.

With the aim of understanding the learning mechanism and dynamics of the masking approaches, other benchmarks were chosen to assess the robustness of the method against the following aspects: discrete and continuous environments; variations across tasks in input, transition and reward distributions. The CT-graph (Soltoggio et al., 2019; 2023) (sparse reward, fast, scalable to large search spaces, variation of reward), Mini-grid (Chevalier-Boisvert et al., 2018; 2023) (variation of input distributions and reward functions) and Continual World (Wołczyk et al., 2021) (continuous robot-control) were used to assess the approaches, with PPO serving as the base RL algorithm. In these benchmarks, the masking approaches ($\text{Mask}_{\text{RI/LC/BLC}} + \text{PPO}$) are compared with a lifelong learning baseline, online EWC multi-head ($\text{EWC}_{\text{MH}} + \text{PPO}$), and with the non-lifelong learning algorithm PPO. Experiments with a PPO single task expert (STE) were also conducted to enable the computation of the forward transfer metric for each method, following the setup in Wołczyk et al. (2021). Control experiments with EWC single head, denoted as $\text{EWC}_{\text{SH}}$, performed poorly as a confirmation that our benchmarks contain interfering tasks (Kessler et al., 2022): we report those results in the Appendix F.1.

The metrics report a lifelong evaluation across all tasks at different points during the lifelong training, computed as the average sum of reward obtained across all tasks in the curriculum. The area under the curve (AUC) is reported in corresponding tables. A forward transfer metric, following the formulation employed in Wołczyk et al. (2021), is computed for the CT-graph, Minigrid and Continual World. For each task, the forward transfer is computed as the normalized difference between the AUC of the training plot for the lifelong learning agent and the AUC for the reference single task expert. For the ProcGen experiments, the lifelong training and evaluation plot format reported in Powers et al. (2022) was followed to enable an easier comparison with the results in the original paper. As tasks are learned independently of other tasks in $\text{Mask}_{\text{RI}}$, there is no notion of forward transfer in the method. Therefore, the method is omitted when forward the transfer metrics are reported.

The results presented in the evaluation plots and the total evaluation metric reported in the tables below were computed as the mean of the seed runs per benchmark, with the error bars denoting the 95% confidence

interval. The *CT8*, *CT12*, *CT8 multi depth*, and *MG10* results contained 5 seed runs per method, while the *CW10* results contained 3 seed runs due to its high computational requirement. While the sample size for the evaluation metric is the number of seed runs, the sample size used for computing the mean and 95% confidence intervals for the forward transfer metric and the training plots is the number of seeds multiplied by the number of tasks per curriculum (i.e., 40, 60, 40, 50, 30 for *CT8*, *CT12*, *CT8 multi depth*, *MG10*, and *CW10* respectively). A significance test was conducted on the results obtained to validate the performance difference between algorithms. The result of the test is reported in Appendix A.

### 5.1 ProcGen

The experimental protocol employed follows the setup of Powers et al. (2022), with a sequence of six tasks $(0 - \text{Climber}, 1 - \text{Dodgeball}, 2 - \text{Ninja}, 3 - \text{Starpilot}, 4 - \text{Bigfish}, 5 - \text{Fruitbot})$ with five learning cycles. Screenshots of the games are reported in the Appendix (D.4). Several environment instances, game levels, and variations in objects, texture maps, layout, enemies, etc., can be generated within a single task. For each task, agents are trained on 200 levels. However, the evaluation is carried out on the distribution of all levels, which is a combination of levels seen and unseen during training.

IMPALA was used as the base RL optimizer on which we deployed the novel masking methods. The results are reported in Figure 2, following the presentation style used in Powers et al. (2022). The masking methods

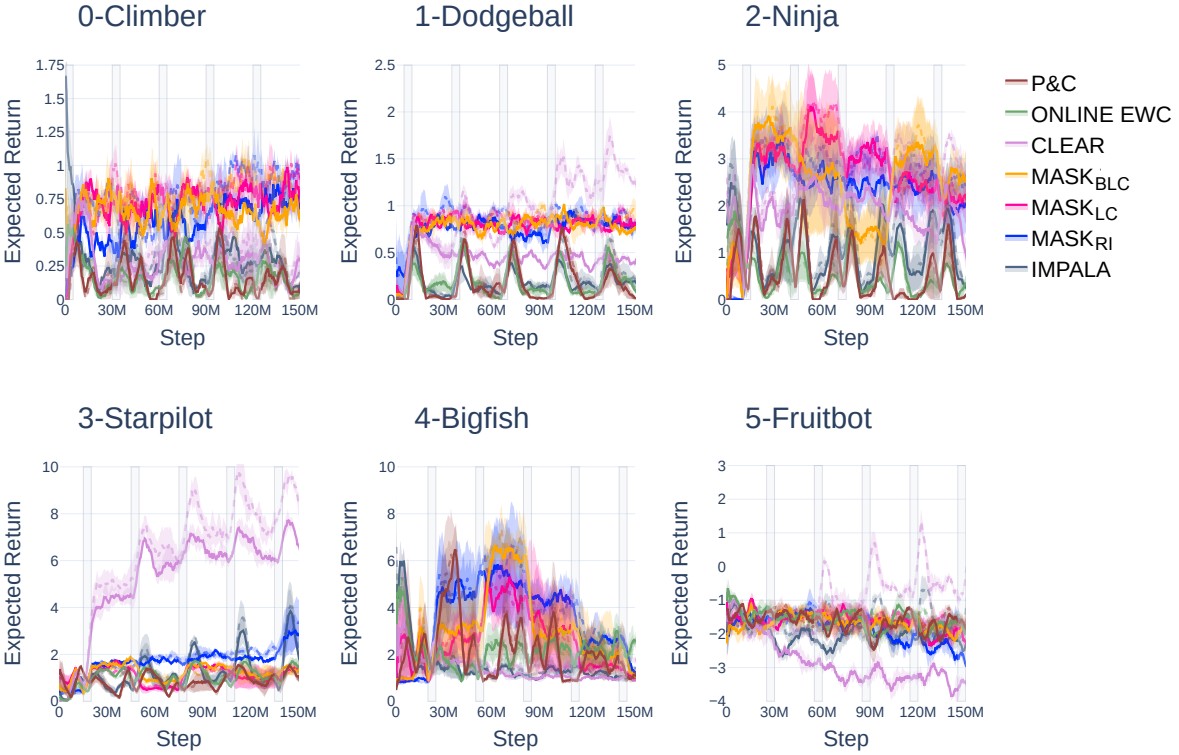

**Figure 2:** Evaluation results in the ProcGen environment (6 tasks, 5 cycles), measured as average across 3 runs. The solid line represents evaluation on unseen environments, while the dotted line represents evaluation on the training environments. The gray shaded rectangles show at what point in time an agent is been trained on each task.

$(\text{Mask}_{\text{RI}}, \text{Mask}_{\text{LC}}, \text{and Mask}_{\text{BLC}})$ show better performance with respect to other baselines across most tasks, while maintaining generalization capabilities across training and evaluation environments. As the tasks are visually diverse, possibly resulting in less similarity across tasks, reusing previous knowledge may not offer much advantage. Nevertheless, the evaluation performance for each method reported in Table 1 illustrates a significant advantage of the masking methods with respect to the baselines, particularly in the test tasks, where $\text{Mask}_{\text{LC}}$ is 44% better than the closest runner up (CLEAR) and over 300% better than P&C.

| | Evaluation Performance | |
| Method | Train Tasks | Test Tasks |
|---|---|---|
| IMPALA | $3687.57 \pm 1194.23$ | $2818.44 \pm 1214.34$ |
| Online EWC | $4645.87 \pm 1963.42$ | $3831.94 \pm 1258.38$ |
| P&C | $3403.59 \pm 3428.37$ | $3390.97 \pm 1795.58$ |
| CLEAR | $11706.34 \pm 5271.45$ | $8447.64 \pm 2286.21$ |
| $\text{Mask}_{\text{RI}}$ | $12462.82 \pm 1057.68$ | $12222.10 \pm 6064.18$ |
| $\text{Mask}_{\text{LC}}$ | $12488.31 \pm 3416.90$ | $12377.77 \pm 1440.80$ |
| $\text{Mask}_{\text{BLC}}$ | $11683.21 \pm 3172.61$ | $11913.48 \pm 3869.95$ |

**Table 1:** Total evaluation return (AUC of the lifelong evaluation plot in Figure 2) on the train and test tasks in ProcGen curriculum. Mean $\pm$ 95% confidence interval reported.

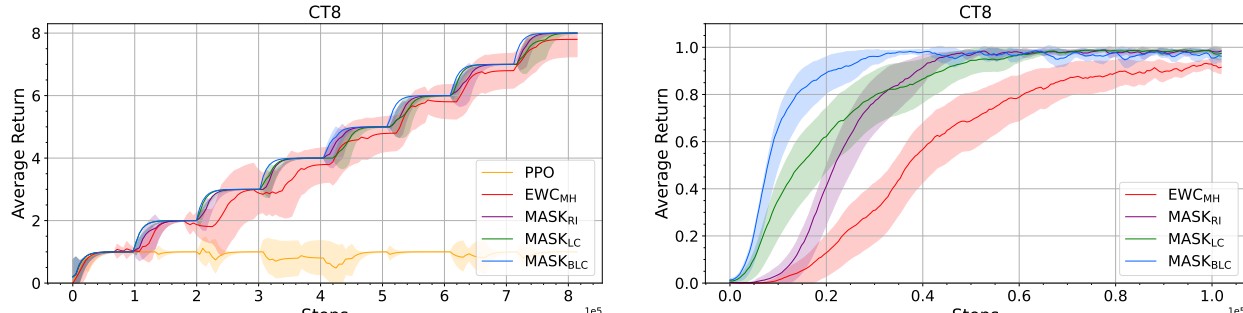

**Figure 3:** Performance in the *CT8* curriculum. (Left) Lifelong evaluation performance on all tasks (mean and 95% confidence interval on 5 seeds/samples). (Right) Training performance on each task, measured as the average return across all tasks and seeds runs (mean and 95% confidence interval on 8 tasks and 5 seeds, i.e., 40 samples). In the lifelong evaluation, a clear difference can be noted between the lifelong learning algorithms and the non-lifelong learning algorithm PPO, which is therefore not reported in the training plot (Right). The numerical values for the AUC are reported in Tables 2. The training curves show how $\text{Mask}_{\text{BLC}}$ is on average faster at learning new tasks, followed by $\text{Mask}_{\text{LC}}$ and $\text{Mask}_{\text{RI}}$. The performance on each individual task is reported in the Appendix in Figure 19

### 5.2   CT-graph

Each instance of the CT-graph environment contains a start state followed by a number of states (2D patterned images) that lead to leaf states and rewards only with optimal policies. A task is defined by setting one of the leaf states as a desired goal state that can be reached only via one trajectory. Two experimental setups were employed: the first with 8 leaf states (reward locations) that serves for 8 tasks, denoted as *CT8* curriculum (depth-3, breadth-2 graph with $2^3 = 8$ leaves). In the second setup, two graph instances with 4 and 8 different reward locations, with depth 2 and 3 respectively, result in combined curriculum of 12 tasks, denoted as *CT12*. Such tasks have levels of similarities due to similar input distributions, but also interfere due to opposing reward functions and policies for the same inputs. Additionally, the 8-task graph has a longer path to the reward that introduces variations in both the transition and reward functions. Graphical illustrations of the CT-graph instances are provided in Appendix D.

Each task is trained for 102.4K time steps. Figures 3 and 4 report evaluations in the *CT8* and *CT12* curricula as agents are sequentially trained across tasks. The forward transfer and the total evaluation performance metrics are presented in Tables 2 and 3 respectively.

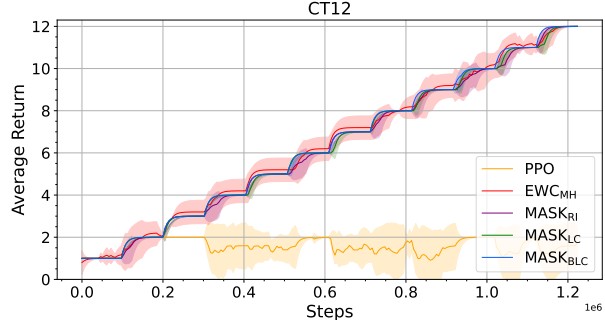 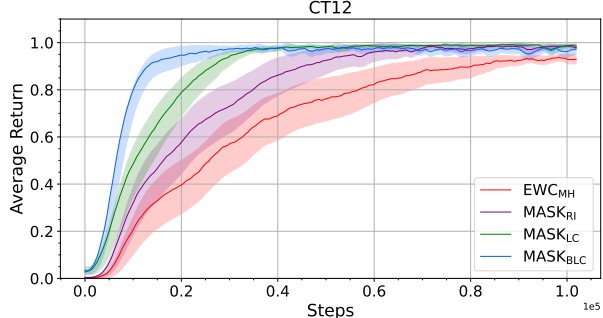

**Figure 4:** Performance in the *CT12* curriculum. (Left) Lifelong evaluation performance on all tasks (mean and 95% confidence interval on 5 seeds/samples). (Right) Training performance on each task, measured as the average return across all tasks and seeds runs (mean and 95% confidence interval on 12 tasks and 5 seeds, i.e., 60 samples).

| Method | Total Eval | Fwd Trnsf. |
|---|---|---|
| PPO | $298.69 \pm 24.12$ | $-0.40 \pm 0.33$ |
| $\text{EWC}_{\text{MH}}$ | $959.38 \pm 141.32$ | $-1.04 \pm 0.32$ |
| $\text{Mask}_{\text{RI}}$ | $1379.71 \pm 51.86$ | – |
| $\text{Mask}_{\text{LC}}$ | $1358.08 \pm 75.29$ | $-0.25 \pm 0.30$ |
| $\text{Mask}_{\text{BLC}}$ | $1405.30 \pm 62.49$ | $0.27 \pm 0.14$ |

**Table 4:** Total evaluation return (AUC of the lifelong evaluation plot in Figure 5(Left)) and forward transfer in the *MG10*. Mean $\pm$ 95% confidence interval reported.

| Method | Total Eval | Fwd Trnsf. |
|---|---|---|
| PPO | $147.00 \pm 13.63$ | $0.15 \pm 0.20$ |
| $\text{EWC}_{\text{MH}}$ | $661.20 \pm 64.14$ | $-0.23 \pm 0.19$ |
| $\text{Mask}_{\text{RI}}$ | $701.20 \pm 6.11$ | – |
| $\text{Mask}_{\text{LC}}$ | $702.00 \pm 5.89$ | $0.46 \pm 0.13$ |
| $\text{Mask}_{\text{BLC}}$ | $716.80 \pm 2.22$ | $0.67 \pm 0.06$ |

**Table 2:** Total evaluation return (AUC of the lifelong evaluation plot in Figure 3(Left)) and forward transfer during lifelong training in the *CT8*. Mean $\pm$ 95% confidence interval reported.

| Method | Total Eval | Fwd Trnsf. |
|---|---|---|
| PPO | $378.40 \pm 22.27$ | $-0.15 \pm 0.18$ |
| $\text{EWC}_{\text{MH}}$ | $1565.00 \pm 92.36$ | $-0.11 \pm 0.14$ |
| $\text{Mask}_{\text{RI}}$ | $1535.00 \pm 10.20$ | – |
| $\text{Mask}_{\text{LC}}$ | $1544.60 \pm 4.17$ | $0.50 \pm 0.08$ |
| $\text{Mask}_{\text{BLC}}$ | $1558.20 \pm 3.09$ | $0.65 \pm 0.04$ |

**Table 3:** Total evaluation return (AUC of the lifelong evaluation plot in Figure 4(Left)) and forward transfer during lifelong training in the *CT12*. Mean $\pm$ 95% confidence interval reported.

The plots show that the masking methods ($\text{Mask}_{\text{RI}}$, $\text{Mask}_{\text{LC}}$, and $\text{Mask}_{\text{BLC}}$) are capable of avoiding forgetting and obtain high evaluation performance, with significantly better forward transfer in comparison to $\text{EWC}_{\text{MH}}$ and PPO. On average, the $\text{Mask}_{\text{LC}}$ approach recovers performance faster than $\text{Mask}_{\text{RI}}$, and $\text{Mask}_{\text{BLC}}$ performs best. An expanded version of the training plots showing the learning curves per tasks and averaged across seed runs is reported in the Appendix F.2.

### 5.3 Minigrid

The experiment protocol employs a curriculum of ten tasks (referred to as *MG10*), which consist of two variants of each of the following: SimpleCrossingS9N1, SimpleCrossingS9N2, SimpleCrossingS9N3, LavaCrossingS9N1, LavaCrossingS9N2. Screenshots of all tasks are reported in the Appendix (D.2). The variations across tasks include change in the state distribution and reward function. The results for the *MG10* experiments are presented in Figure 5 and Table 4. The masking methods obtained better performance in comparison to the baselines, with $\text{Mask}_{\text{BLC}}$ obtaining the best performance. Appendix F.2 provides a full experimental details.

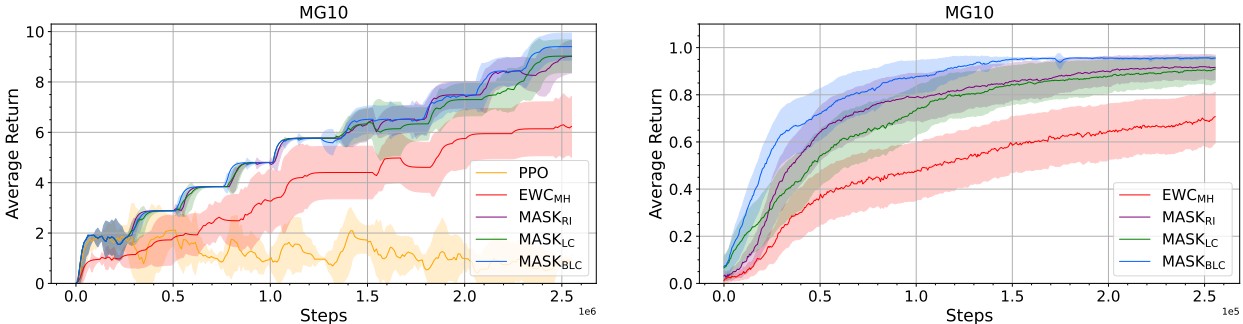

**Figure 5:** Performance in the *MG10* curriculum. (Left) Lifelong evaluation performance on all tasks (mean and 95% confidence interval on 5 seeds/samples). (Right) Training performance on each task, measured as the average return across all tasks and seeds runs (mean and 95% confidence interval on 10 tasks and 5 seeds, i.e., 50 samples).

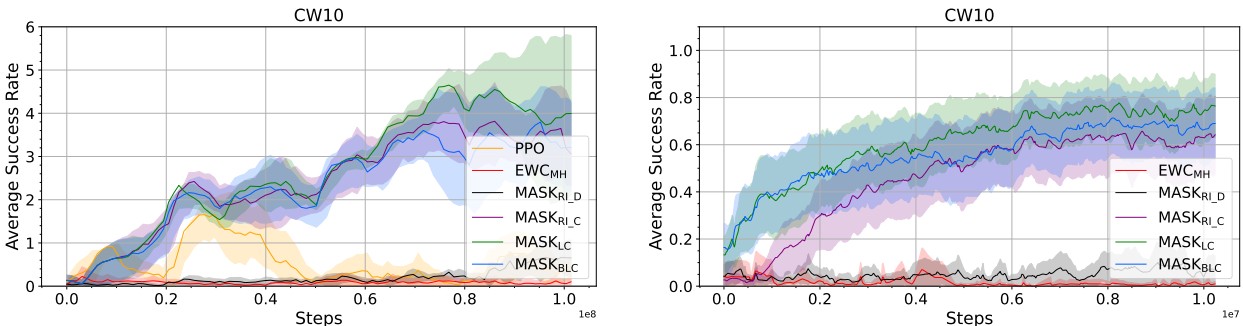

**Figure 6:** Performance in the *CW10* curriculum measured using the success rate metric. (Left) Lifelong evaluation performance on all tasks (mean and 95% confidence interval on 3 seeds/samples). (Right) Training performance on each task, measured as the average success rate across all tasks and seeds runs (mean and 95% confidence interval on 10 tasks and 3 seeds, i.e., 30 samples).

## 5.4 Continual World

We evaluated the novel methods in a robotics environment with continuous action space, the Continual World (Wołczyk et al., 2021). The environment was adapted from the MetaWorld environment (Yu et al., 2020) that contains 50 classes of robotics manipulation tasks. The CW10 curriculum consists of 10 robotics tasks: visual screenshots are provided in the Appendix (D.3). The results for all methods were measured using the success rate metric introduced in Yu et al. (2020), which awards a 1 if an agent solves a task or 0 otherwise. For the masking methods in this curriculum, the standard quantization of masks into binary performs poorly. To demonstrate this, two variants are run: the standard setting, where a binary mask is derived from the scores, denoted as $\text{Mask}_{\text{RI\_D}}$, and another where a continuous mask is derived from the scores (discussed in Section 4.3), denoted as $\text{Mask}_{\text{RI\_C}}$. The results from Figure 6 and Table 5 show that $\text{Mask}_{\text{RI\_C}}$ performs significantly better than $\text{Mask}_{\text{RI\_D}}$. Motivated by the results, the linear combination of masks method $\text{Mask}_{\text{LC}}$ presented for this curriculum also employed the use of continuous masks. $\text{Mask}_{\text{LC}}$ and $\text{Mask}_{\text{BLC}}$ performed markedly better than the baseline $\text{EWC}_{\text{MH}}$ that appears to struggle on this benchmark. Appendix F.2 reports the details for all methods.

## 6 Analysis

The results of the previous section prompt the following questions: what linear coefficients emerge after learning? How is rapid learning in $\text{Mask}_{\text{LC}}$ achieved? How is knowledge reused?

| Method | Total Eval | Fwd Trnsf. |
|---|---|---|
| PPO | $53.83 \pm 72.32$ | $-4.06 \pm 2.58$ |
| $\text{EWC}_{\text{MH}}$ | $8.60 \pm 12.91$ | $-7.39 \pm 3.76$ |
| $\text{Mask}_{\text{RI\_D}}$ | $20.45 \pm 51.46$ | – |
| $\text{Mask}_{\text{RI\_C}}$ | $246.83 \pm 157.39$ | – |
| $\text{Mask}_{\text{LC}}$ | $272.43 \pm 124.92$ | $-0.33 \pm 0.36$ |
| $\text{Mask}_{\text{BLC}}$ | $237.00 \pm 167.25$ | $-0.61 \pm 0.52$ |

**Table 5:** Total evaluation success metric (AUC of the lifelong evaluation plot in Figure 6(Left)) and forward transfer in the *CW10*. Mean $\pm$ 95% confidence interval reported.

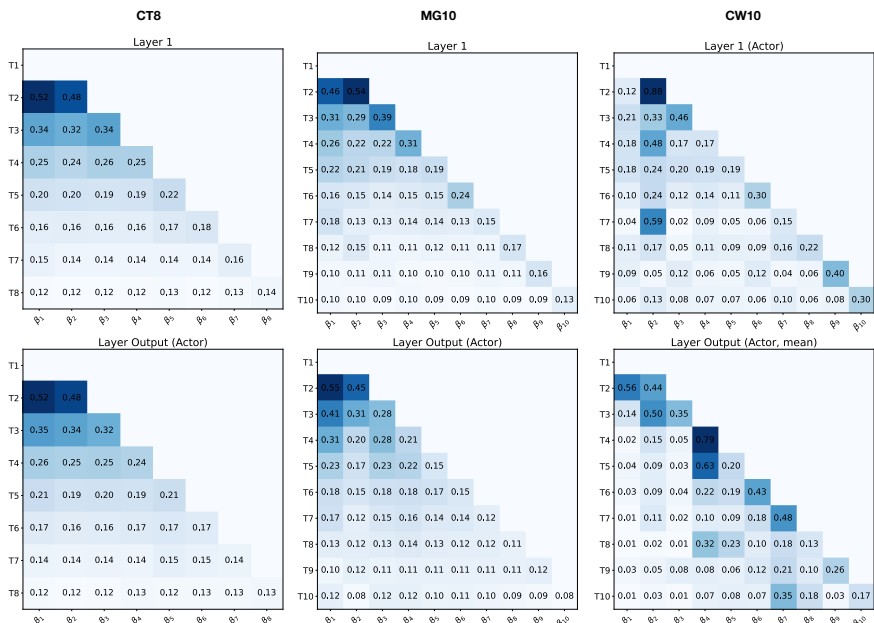

**Figure 7:** Coefficients $\bar{\beta}$ in $\text{Mask}_{\text{LC}}$ after training on the *CT8*, *MG10*, and *CW10* curricula. Each row represents a task, and the values (which sums to 1) in it are the final set of coefficients after training on the task: the higher the value of a cell, the higher the level of importance of the corresponding mask in the linear combination operation. The figure only shows coefficients for the first layer and the output (actor) layer. See Appendix E.2 for plots of all other layers.

Another interesting question that arise is the training efficiency derived from knowledge reuse in mask methods. The training plots in Figures 3(Right), 4(Right), 5(Right), and 6(Right) showed that the mask methods required fewer training steps on average to learn tasks in comparison to the baselines. For interested readers, Appendix H reports the analysis conducted on the time (i.e., training steps) taken to reach certain level of performance per task.

## 6.1 Coefficients for the linear combination of masks

To validate whether the proposed linear combination process can autonomously discover important masks that are useful for the current task, a visualization of the co-efficients after learning each task is presented.

Figure 7 presents the visualization of the coefficients (for the input and output layers) for a $\text{Mask}_{\text{LC}}$ agent trained in the *CT8*, *MG10*, and *CW10* curricula respectively. In each plot, each row reports the final set of coefficients after training on a task. For example, the third row in each plot represents the third task in the curriculum and reports three coefficients used for the linear combination of the masks for two tasks and the new mask. For the first task (first row), there are no previous masks to combine.

For the *CT8* and *MG10*, the plots show that the coefficients have similar weights for each task (i.e., row wise in Figure 7). This observation is consistent across the layers of the network (see Appendix E.2 for plots across all layers). The uniform distribution across co-efficients means that the knowledge from all previous tasks is equally important and reused when learning new tasks, possibly indicating that tasks are equally diverse or similar to each other. The knowledge reuse of previous tasks thus accelerates learning and helps the agent quickly achieve optimal performance for the new task. For example, in the CT-graph curricula where navigation abilities are essential to reach the goal, the knowledge on how to navigate/traverse to different parts of the graph is encoded in each previously learned task. Rather than re-learn how to navigate the graph in each task, and subsequently the solve the task, the agent can leverage on the existing navigational knowledge. The performance improvement therefore comes from the fact the agent with knowledge reuse can leverage existing knowledge to learn new tasks quickly. Note that we will not expect any performance improvement if the new task bears no similarity with the previously learned task.

Comparing the values across layers (i.e., column wise in Figure 7), we note that there is little variation. In other words, the standard deviation of each vector $\bar{\beta}$ is low, as all values are similar. From such an observation, it follows that the vector $\bar{\beta}$ could be replaced by a scalar for these two benchmarks.

A different picture emerges from the analysis of the coefficients in the CW10 curriculum (Figure 7 rightmost column). Here the coefficients appear to have a larger standard deviation both across masks and across layers. Particular values may suggest relationships between tasks. E.g., the input layer coefficient $\beta_2^1$ (from layer 1, mask 2) is high in task 4 and 7. Similar diverse patterns can be seen in the output layer. We speculate that tasks in CW10 are more diverse and the optimization process is enhancing specific coefficients to reuse specific skills. The non-uniformity of the co-efficients in the analysis could be a consequence of different levels of task similarities as reported in the forward transfer matrix in Wołczyk et al. (2021) for the CW10.

## 6.2 Exploitation of previous knowledge

The linear combination of masks appears to accelerate learning significantly as indicated in Figures 3(Right), 4(Right), 5(Right), and 6(Right). To investigate the causes of such learning dynamics, we plot the probabilities of actions during an episode in a new task. The idea is to observe what are the softmax probabilities that are provided by the linear combination of masks at the start of learning for a new task. A full analysis would require the unfeasible task of observing all trajectories: we therefore focus on the optimal trajectory by measuring probabilities when traversing such an optimal trajectory.

Figure 8 shows the analysis for the following cases: facing a 4th task after learning 3 tasks in the *CT8* benchmark; facing the 12th task after learning 11 tasks in the *CT12* benchmark; facing the 7th task after learning 6 tasks in the *MG10* benchmark. The chosen task for each curriculum is set to test different instances of knowledge reuse. For *CT8*, the 4th task tests the agent's ability to reuse knowledge after learning a few similar tasks, while the 12th task in *CT12* investigates the agent's behavior after learning many tasks. In *MG10*, the input distribution of the 7th task differs from the previous ones (i.e., from tasks with no lava to tasks with lava), thus testing the agent's ability to reuse previously learned navigation skills while dealing with new input scenarios. The result of the analysis is generalizable to other task changes in the curricula.

In the analysis, $EWC_{MH}$ produced a purely random policy, with a uniform distribution over actions for each time step. Despite learning previous tasks, the new random head results in equal probabilities for all actions. On the contrary, both $Mask_{LC}$ and $Mask_{BLC}$ use previous masks to express preferences. In particular, in the *CT8* and *CT12*, the policy at steps 1, 3, 5 and 7 coincides with the optimal policy for the new task, likely providing an advantage. However, at steps 4 and 6 for the *CT8*, and 2, 4, and 6 for the *CT12*, $Mask_{LC}$ has a markedly wrong policy: this is due to the fact that the new task has a different reward function and is therefore interfering. Due to the balanced combination of previous knowledge with a new mask, $Mask_{BLC}$ seems to strike the right balance between trying known policies and exploring new ones. Such a balanced approach is also visible in the *MG10* task. Here, task 7 (see the Appendix D.2) consists of avoiding the walls and the lava while proceeding ahead, then turning left and reaching the goal: most such skills are similar to those acquired in previously seen tasks, and therefore task 7 is learned starting from a policy that is close to being optimal.

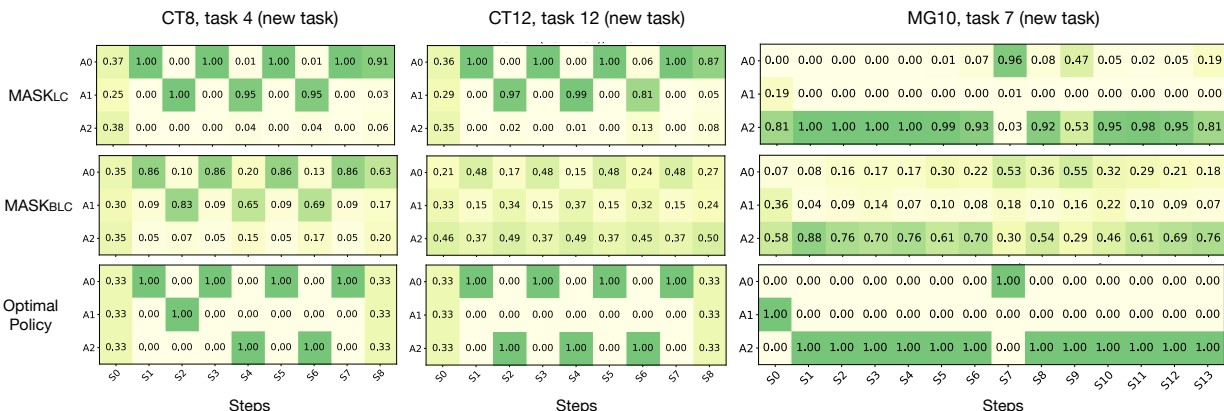

**Figure 8:** Knowledge reuse when learning a new tasks. The softmax output probabilities are shown during an episode with an unforeseen task (task 4 in *CT8*, task 12 in *CT12*, and task 7 in *MG10*) as the agent is guided through an optimal policy. From the top row down, Mask$_{LC}$ displays biased probabilities representing an average behavior across previous tasks. Mask$_{BLC}$ shows biased probabilities but contains more randomness in comparison to Mask$_{LC}$. A visual comparison with the optimal policy (bottom row) suggests that both Mask$_{LC}$ and Mask$_{BLC}$ start learning unforeseen tasks with useful knowledge. Note, EWC$_{MH}$ produced balanced probabilities (uniform distribution) for all actions at each time step, and is thus omitted from the figure.

If Mask$_{LC}$ and Mask$_{BLC}$ are capable of exploiting previous knowledge, it is natural to ask whether such knowledge can be exploited to learn increasingly more difficult tasks. The CT-graph (Soltoggio et al., 2019) environment allows for increasing the complexity by increasing the depth of the graph. In particular, with a depth of 5, the benchmark results in a highly sparse reward environment with a probability of getting a reward with a random policy of only one in $3^{11} = 177,147$ episodes (Soltoggio et al., 2023). We therefore designed a curriculum, the *CT8 multi depth*, composed of a set of related but increasingly complex tasks with depth 2, 3, 4 and 5 (two tasks each depth with a different reward function).

Figure 9 shows the performance in the *CT8 multi depth* curriculum. EWC$_{MH}$ was able to learn the first 4 tasks with depth 2 and 3, but failed to learn the last 4 more complex tasks. Interestingly, Mask$_{BLC}$ managed to learn task 5 and 6 only partially. Mask$_{LC}$ was able to learn all tasks, demonstrating that it could reuse previous knowledge to solve increasingly difficult tasks. Figure 10 presents the training performance for each individual task, highlighting were most methods fail in the curriculum.

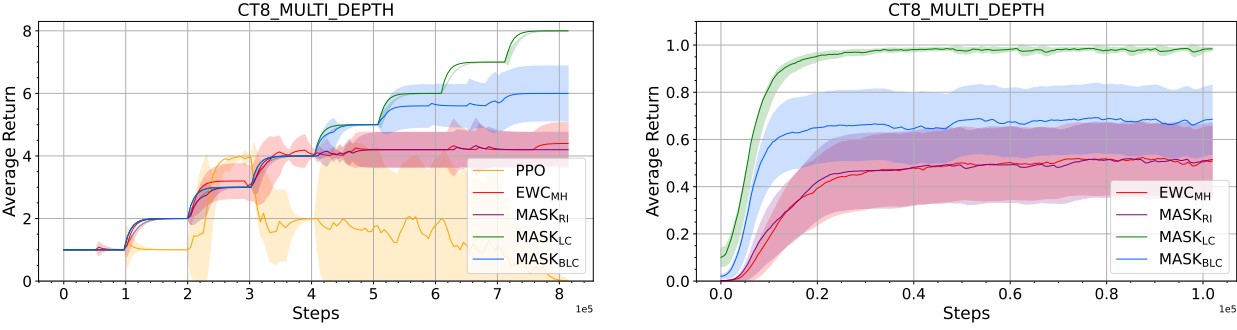

**Figure 9:** Performance in the *CT8 multi depth* curriculum. (Left) Lifelong evaluation performance on all tasks (mean and 95% confidence interval on 5 seeds/samples). (Right) Training performance on each task, measured as the average return across all tasks and seeds runs (mean and 95% confidence interval on 8 tasks and 5 seeds, i.e., 40 samples). Excluding MASK$_{LC}$, the training performance for other methods are sub-optimal, due to the failure to solve later tasks in the curriculum.

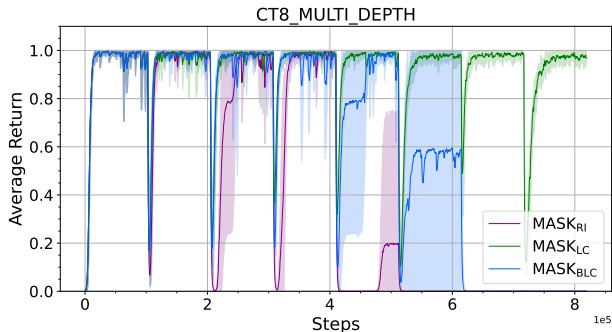

**Figure 10:** Training performance on each task, measured as the average return across all seeds runs (mean and 95% confidence interval on 5 seeds/samples), expanded from Figure 9(Right).

| Method | Total Eval | Fwd Trnsf. |
|---|---|---|
| PPO | $237.80 \pm 115.65$ | $0.05 \pm 0.11$ |
| $EWC_{MH}$ | $540.00 \pm 39.88$ | $0.10 \pm 0.08$ |
| $Mask_{RI}$ | $533.00 \pm 41.13$ | – |
| $Mask_{LC}$ | $719.60 \pm 0.68$ | $0.79 \pm 0.06$ |
| $Mask_{BLC}$ | $644.00 \pm 41.47$ | $0.48 \pm 0.11$ |

**Table 6:** Total evaluation return (AUC of the lifelong evaluation plot in Figure 9(Left)) and forward transfer during lifelong training in the *CT8 multi depth*. Mean $\pm$ 95% confidence interval reported.

## 7 Discussion

The proposed approach employs learned modulatory masking methods to deliver lifelong learning performance in a range of reinforcement learning environments. The evaluation plots (the left panels of Figures 3, 4, 5 and 6) show how the performance increases as the agent learns through a given curriculum. The monotonic increase, indicating minimal forgetting, is most clear for the CT-graph and Minigrid, while the Continual World appears to have a more noisy performance. The $EWC_{MH}$ baseline algorithm shows to be a highly performing baseline in the *CT8* and *CT12* curricula, it performs less well in the MG10 curriculum, and poorly in the *CW10* curriculum. The masking methods, instead, perform consistently in all benchmarks, with the linear combination methods, $Mask_{LC}$ and $Mask_{BLC}$, showing some advantage over the random initialization $Mask_{RI}$. Evaluations on the ProcGen environments, while noisy to interpret from Figure 2, reveal that the masking methods outperform IMPALA, Online EWC, P&C and CLEAR by significant margins (Table 1).

While the learning dynamics of the core algorithm $Mask_{RI}$ indicate superior performance to the baselines, we focused in particular on two extensions of the algorithm, $Mask_{LC}$ and $Mask_{BLC}$. These use a linear combination of previously learned masks to search for the optimal policy in a new unforeseen task. These two variations combine previously learned masks with a new random mask to search for the optimal policy. One catch with this approach is that the performance on a new task will depend on which and how many tasks were previously learned. However, this is a property of all lifelong learning algorithms that leverage on previous knowledge. The balanced approach $Mask_{BLC}$ starts with a 0.5 weight on the new random mask and can be seen as a blend between the random initialization $Mask_{RI}$ and the linear combination $Mask_{LC}$. On average, it appears to achieve slightly better performance than either $Mask_{RI}$ or $Mask_{LC}$. The standard linear combination $Mask_{LC}$ was the only algorithm able to learn the most difficult task on the CT-graph. This suggests that the algorithm is capable of exploiting previous knowledge to solve challenging RL problems. The fact that both $Mask_{LC}$ and $Mask_{BLC}$ have superior performance to the core random initialization $Mask_{RI}$ validates the hypothesis that previous knowledge stored in masks can be reused.

The analysis of the coefficients of the linear combination (Section 6.1) reveals that the optimization can tune them to adapt to the nature of the curriculum. In the CT-graph and Minigrid curricula, previous masks are

used in balanced proportions. On the Continual World environment, instead, particular masks, and layers within those masks, had significantly larger weights than others. From this observation, we conclude that the new proposed approach may be flexible enough to adapt to a variety of different curricula with different degrees of task similarity.

One concern with modulating masks is that memory requirements increase linearly with the number of tasks. This makes the approach not particularly scalable to large numbers of tasks. However, the promising performance of the linear combination approaches suggests that an upper limit could be imposed on the number of masks to be stored. After such a limit has been reached, new tasks can be learned solely as linear combinations of known masks, significantly reducing memory requirements. While this paper tested vector parameters with a scalar for each network layer, the analysis of the tuned parameters suggests that in some cases a single scalar for each mask could be used, further reducing memory requirements. Other suggestions to combat memory requirements in the mask approach are discussed in Appendix G.

In the modulating masking setup, it is assumed that a task oracle informs the lifelong RL agent of task boundaries and the task presented at any given time (during training and evaluation/testing). This is made possible by providing a task identifier to the agent to select the correct mask. While the explicit specification of task boundaries is a limitation of all LRL methods that make this assumption, the nature of the proposed mask method, where a mask is associated to a task, implies that existing task detection methods could be combined with the mask setup to address this limitation. One approach could involve the use of *forget-me-not* Bayesian approach to task detection (Milan et al., 2016), as was employed in Kirkpatrick et al. (2017). Another approach that could be explored is the use of optimal transport methods (Alvarez-Melis & Fusi, 2020; Liu et al., 2022) to measure distance of states/input across tasks. Also, the few-shots optimization of the linear superposition of masks via gradient descent that was employed in Wortsman et al. (2020) for LSL could be employed to infer task mask in the LRL setup.

Given the fixed nature of the backbone network and the use of binary masks to sparsify the network, the representational capacity of the network could be affected. Masking approaches have been extensively studied in fixed neural networks (Zhou et al., 2019; Ramanujan et al., 2020), including lifelong supervised learning setup (Mallya et al., 2018; Wortsman et al., 2020), with discussions about representational capacity and generalization. While representation capacity could be affected, there are gains in generalization Zhou et al. (2018); Frankle & Carbin (2018) and robustness to noise (Arora et al., 2018) in sparse networks. In the future investigations, the gains in generalization could be useful to model-based RL approaches in lifelong learning.

## 8 Conclusion

This work introduces the use of modulating masks for lifelong reinforcement learning problems. Variations of the algorithm are devised to ignore or exploit previous knowledge. All versions demonstrate the ability to learn on sequential curricula and show no catastrophic forgetting thanks to separate mask training. The analysis revealed that using previous masks to learn new tasks is beneficial as the linear combination of masks introduces knowledge in new policies. This finding promises potential new developments for compositional knowledge RL algorithms. Exploiting previous knowledge, one version of the algorithm was able to solve extremely difficult problems with reward probabilities as low as $5.6 \cdot 10^{-6}$ per episode simply using random exploration. These results suggest that modulating masks are a promising tool to expand the capabilities of lifelong reinforcement learning, in particular with the ability to exploit and compose previous knowledge to learn new tasks.

## Broader Impact Statement

The advances introduced in this work contribute to the development of intelligent systems that can learn multiple tasks over a lifetime. Such a system has the potential for real-world deployment, especially in robotics and automation of manufacturing processes. As real-world automation increases, it may lead to reduce the demand for human labor in some industries, thereby impacting the economic security of people. Also, when such systems are deployed, careful considerations are necessary to ensure a smooth human machine collaboration in the workforce, ethical considerations and mitigation of human injuries.

## Acknowledgments

This material is based upon work supported by the United States Air Force Research Laboratory (AFRL) and Defense Advanced Research Projects Agency (DARPA) under Contract No. FA8750-18-C-0103 (Lifelong Learning Machines) and Contract No. HR00112190132 (Shared Experience Lifelong Learning). Any opinions, findings and conclusions or recommendations expressed in this material are those of the author(s) and do not necessarily reflect the views of the United States Air Force Research Laboratory (AFRL) and Defense Advanced Research Projects Agency (DARPA).

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
