# OpenReview forum: "Lifelong Reinforcement Learning with Modulating Masks"
_TMLR — Accepted by TMLR_

### Review · Reviewer_Cewb · 2023-02-13

**Summary Of Contributions:**

The authors experiment with a sparse masking approach to Continual Learning, specifically in the RL context. In this setting, each task is represented by a sub-network indexed through a gating masks that are adapted online. Different from previous settings, the authors exclusively work in the setting of unchanging weights and thus no forgetting. The contributions of this work are:

- A new mask initialization scheme which facilities the re-use of skills encoded in previous tasks through a linear combination of the respective sub-masks.
- A large, empirical evaluation of the proposed changes.

**Audience:**

No

**Broader Impact Concerns:**

No concerns

**Claims And Evidence:**

Yes

**Requested Changes:**

Requested changes:

Abstract:
- "learning multiple input distributions, typically in classification". Classification does not usually involve learning the input distribution (i.e. modelling p(x)), please rephrase.
- "PPO/IMPALA" - Citation missing.
- Please mention here that you're making the assumption of fixed weights.

Introduction:
- What is the difference between LRL and Continual Learning? Why the insistence that this is a separate field ("LRL shares some of the objectives of CL [...] but [...] has developed into a separate field")? This does not seem like a commonly accepted categorization to me. Indeed, the most of the techniques evaluated in the experimental section (e.g. EWC, CLEAR, P&C) as well as the technique proposed by the authors would be expected to be applicable to other CL tasks.
- "This study introduces the use of modulating masks in LRL" - Incorrect claim, modulating masks have been used before. (e.g. PackNet, [1])
- "DQN [...] EWC to produce a lifelong learning DeepRL agent." - Incorrect, the original EWC paper already showed experiments on Deep RL.

Related Work:
- "for example, DQN, PPO, or SAC" - citations missing for PPO/SAC.
- Missing reference for [1] (which shows modulating masks on Deep RL), [2, 3] for Continual RL in general. [4, 5] for alternative uses of modulating masks in CL.  Some of [1, 2, 3] should be used as baselines.

Background:

- 3.1: Define what expectation is over when writing the  expected cumulative discounted reward.
- 3.2: Define sigmoid. - Please mention iterative magnitude pruning as a default way of implementing sparse NN. - Also useful to cite a review paper (e.g. [6]).

Methods:
- 4.2: "we instead bootstrap the learning from the start" - Not sure what the meaning of "bootstrap" is in this context. - Why is the superscript $r$ used in the definition of $\phi^r$? Wouldn't $k+1$ make more sense? - $\bar{\beta}$ undefined (Is this the initaliztation?).
Algorithm 1, Line 10: This should be an assignment, not an equal sign.

Experiments:
- "As tasks are learned independently of other tasks in MaskRI, there is no notion of forward transfer in the method" - This is a direct consequence and drawback of not updating the weights.

Analysis:
- Figure 7 (left, middle) If MASK_LC resorts to a uniform distribution, where does the performance improvement come from?
- Figure 7 (right) "We speculate that tasks in CW10 are more diverse and the optimization process is enhancing specific coefficients to reuse specific skills." This should be analyzed more thoroughly.
- "shows the analysis for the following cases: facing a fourth task after learning three tasks in the CT8benchmark; facing the 12th task after learning 11 tasks in the CT12 benchmark; facing the 7th task after learning 6 tasks in the MG10" - Why are such specific numbers chosen? Does the same finding hold for all other combinations or this is a result of cherry picking?
- "[...] validates the hypothesis that previous knowledge stored in masks can be reused." - I'm not necessarily convinced given the uniform Finding in Figure 7. A more thorough analysis would be useful here. What aspects of those previous tasks are useful?
- Figure 8: No need to show the result for EWC. Simply state it's a uniform distribution.


Discussion:
- There is too much emphasis on memory costs in my opinion. It is easy to see that the memory cost compares very favorably to commonly used techniques based on replay so this is unlikely to be a major concern. This could be moved to the appendix.
- Please discuss the points raised above (task-inference, untrained weights).


[1] Schwarz, Jonathan, et al. "Powerpropagation: A sparsity inducing weight reparameterisation." Advances in neural information processing systems 34 (2021): 28889-28903.

[2] Kaplanis, Christos, Murray Shanahan, and Claudia Clopath. "Continual reinforcement learning with complex synapses." International Conference on Machine Learning. PMLR, 2018.

[3] Kaplanis, Christos, Murray Shanahan, and Claudia Clopath. "Policy consolidation for continual reinforcement learning." arXiv preprint arXiv:1902.00255 (2019).

[4] Sokar, Ghada, Decebal Constantin Mocanu, and Mykola Pechenizkiy. "Spacenet: Make free space for continual learning." Neurocomputing 439 (2021): 1-11.

[5] Von Oswald, Johannes, et al. "Learning where to learn: Gradient sparsity in meta and continual learning." Advances in Neural Information Processing Systems 34 (2021): 5250-5263.

[6] Hoefler, Torsten, et al. "Sparsity in deep learning: Pruning and growth for efficient inference and training in neural networks." The Journal of Machine Learning Research 22.1 (2021): 10882-11005.

**Strengths And Weaknesses:**

Strengths:
- Experimental evaluation is extensive, providing results on various complex reinforcement learning environments.
- Some interesting attempts at analysis. Would prefer this section to be expanded.
- I enjoyed this discussion around biological learning (Section 2.3)

Weaknesses:
- Limited algorithmic novelty. CL based on sparse gating is a well known technique, the idea to linearly combine previous masks is relatively straight-forward.
- The authors fail to discuss some of the inherent questions and concerns of the proposed approach, specifically:
     - Given that the weights are randomly initialized and fixed (critical to avoid forgetting), how much representational capacity is lost? What are the limitations of learning purely based on gating a random network? To which extent does this depend on size and complexity of the backbone network?
     - The method requires task-inference at test time (so the correct score parameters $S_k$ can be chosen for task $k$). How do the authors propose to infer the task in a setting where the agent is evaluated on an unknown but previously encountered problem?
- Related Work section should be improved. The authors miss several relevant works in Continual RL / CL based on Sparsity.
- Writing and presentation could be improved. Several crucial facts should be stated more clearly. The paper is unnecessarily long.
- Baseline methods chosen for evaluation are relatively simple and known to be significantly under-performing SOTA. No comparison to another technique based on sub-networks despite results being readily available (e.g. PackNet on CW10).


Overall unfortunately not yet ready for publication.

---

> ### Author Response · Authors · 2023-03-24
> **Response to Reviewer Cewb (1/4)**
>
> We thank the reviewer for the insightful and useful comments. Please see our response to the feedback below. All the requested changes have been addressed in the revised version of paper.
>
> Weaknesses
> 1. **Limited algorithmic novelty:** We agree that the algorithmic method to linearly combine masks is relative straightforward. However, whether the method enables knowledge reuse with different and random growing curricula of RL tasks was still an open question. It was also unclear which specific initial parameter configuration is ideal. The possible advantages were unknown particularly when tasks have different degrees of similarities, grow in complexity, and may also be interfering with one another.  Our work is the first to address these unknowns and show the effectiveness of linear combinations of learned masks in RL. A paragraph to make this point clearer in the paper was modified in the introduction.
> 2. **Representational capacity lost due to fixed network, network size and performance:**
> That is an interesting comment about learned modulating mask in general. Learned masks have been extensively studied in fixed neural networks [5, 6], including lifelong supervised learning setup [1, 7], with discussions about representational capacity and generalization in sparse networks derived from masks. While representation capacity could be lost due to sparsity, there are gains in generalization [3, 4] and robustness to noise [2]. Building on these earlier studies, we focused our investigation on the use of learned masks in lifelong RL, and also introduced forward transfer in the mask setup. An extensive study of representations from sparse networks in lifelong RL could be a potential direction to explore in the future, as it could be potentially useful in model-based RL approaches. The above text has been added to the revised paper in Section 7.
> 3. **Task inference at test time:**
> To address this comment, we have added the text below to Section 7 of the revised paper.
> In the current setup, it is assumed that a task oracle informs the agent of which task is presented at any given time (during training and evaluation/testing). This is made possible by providing a task id to the agent. Several solutions could be devised to eliminate this assumption in the system. One approach could involve the use of Bayesian approach to task detection [8] as was done in [9]. Another approach that could be explored is the use of optimal transport methods [10, 11] to measure distance of states/input across tasks.
> 4. **Missing citations in related works:**
> Thanks for highlighting the missed literatures. We have added them [13 – 18] in Section 2.2 of the revised paper. We’ll be happy to consider adding additional references if the reviewer has specific papers in mind.
> 5. **Improvement in writing and presentation style:**
> We have made an effort to improve the readability and reduce the paper length (in Sections 1, 2 and 5) to the best of our capacity. However, if there are specific sections in the paper that could be further reduce, we would be happy to address those and perhaps move them to the supplementary material.
> 6. **Baselines (e.g., policy consolidation and complex synapse time scale):**
> Well-known baselines (P&C, Online EWC, CLEAR, and IMPALA) are used on the ProcGen environment for overall performance comparisons. However, for the specific analysis of learned masks and forward transfer in LRL, rather than demonstrating absolute SoTA, we focused on the different behaviors of the suggested variations of the proposed method.
> 7. **Comparison against PackNet CW10 public result (from the Continual World paper):**
> The publicly available PackNet result for CW10 cannot be used a baseline for two reasons: (i) the underlying RL algorithm and training setup are different from the ones used in this work (e.g., off-policy SAC algorithm was employed in the PackNet result, while we used on-policy PPO algorithm), and (ii) the CW10 results published in the Continual World paper were based on the v1 meta-world environments which have been reported to contain buggy reward function (see links below), instead, we used the v2 environments which contain the updated reward functions (Appendix D.3 of the paper reports the use of v2 meta-world environments). Discussion links: https://github.com/rlworkgroup/metaworld/issues/226, https://github.com/awarelab/continual_world/issues/2

---

> > ### Comment · Reviewer_Cewb · 2023-05-18
> > **Response to authors**
> >
> > Thank you to the authors for taking the time to formulate a response and incorporating suggested references to related work and other writing improvements.
> >
> > Unfortunately, I am disappointed by the lack of concrete algorithmic and experimental improvements provided by the authors. While I understand that not all criticisms and changes can necessarily be integrated in the available time, the authors seem hesitant to address any of my concerns, be that (i) Task inference at test time (merely suggesting existing possible solutions without testing them), (ii) additional baselines (arguing that they needn't be used), (iii) very closely related work e.g. PackNet on CW10 (which even if the available result is not applicable can be easily re-implemented and run in a setup equivalent to the author's results) or (iv) improving representational capacity by introducing one of many available network growth techniques. Unfortunately, these responses give an overall impression of a missed opportunity to add substantial improvements to the method proposed.

---

> > > ### Author Response · Authors · 2023-05-22
> > > **Response to Reviewer Cewb**
> > >
> > > Dear Reviewer Cewb,
> > >
> > > Thank you for response. We believed we addressed all your comments to the best our ability. Nonetheless, our sincere apologies if you believe they were not sufficiently addressed.
> > >
> > > **(i) Task inference at test time (merely suggesting existing possible solutions without testing them)**
> > > The reviewer’s initial comment requested a discussion about the task inference: We provided that in the revised paper, and pointed that out in point 3 of our previous response https://openreview.net/forum?id=V7tahqGrOq&noteId=yu8V2d4WLL
> > > We infer from the latest comment that the reviewer would like to see experimental evidence of task inference in masks: we believe this is not too aligned with the paper’s objectives, but we can implement and test the task inference methods from Wortsman et al. (2020) and provide the analysis in the final version of the paper.
> > >
> > >
> > > **(ii) additional baselines (arguing that they needn't be used)**
> > > We ran a set of baselines (4 baselines including EWC, P&C, CLEAR and IMPALA) that we considered most appropriate, and stress that the aim of the study is not to demonstrate absolute SoTA on performance. While considering that each baseline comes at an implementation and computational cost, we are happy to consider specific additional suggestions if they strengthen the paper’s claims, i.e., that learned masking methods are promising for lifelong RL domains and knowledge reuse in masks.
> > >
> > >
> > > **(iii) very closely related work e.g. PackNet on CW10 (which even if the available result is not applicable can be easily re-implemented and run in a setup equivalent to the author's results)**
> > > The CW10 experiments have a high computational demand, taking approximately 3 days (per seed) to run on our GPU machine, in addition to the time to re-implementation of PackNet in our setup. The goal of our tests is not to show that our approach beats other masking approaches, but rather investigate how the mask variants perform in continuous action space. As we cannot see how the suggested experiment supports the claims of the paper, we unfortunately choose not to follow this recommendation. Nevertheless, we do share the curiosity of the reviewer and agree that if we had large computational resources and engineering time at disposal, we would investigate the performance of more related methods.
> > >
> > >
> > >  **(iv) improving representational capacity by introducing one of many available network growth techniques.**
> > > We note that this is an important topic in masking methods, and we revised the paper to point out studies that have specifically focused on this aspect. However, introducing or investigating methods to improve representational capacity is not an objective of this study: in our paper revision, we improved the text to better explain the objectives of the paper.
> > >
> > > Kind regards,
> > > Authors
> > >
> > >
> > > **References**
> > > [1] Wortsman, M., Ramanujan, V., Liu, R., Kembhavi, A., Rastegari, M., Yosinski, J. and Farhadi, A., 2020. Supermasks in superposition. Advances in Neural Information Processing Systems, 33, pp.15173-15184.

---

### Review · Reviewer_JcNN · 2023-03-24

**Summary Of Contributions:**

This paper presents a modulation based lifelong learning approach in the RL domain. It describes different schemes to learn masking scores for each task, including Mask Random Initialization, Mask Linear Combination and their hybrid Balanced Linear Combination. All these schemes are proposed to well leverage the previously trained masking score, so as to enhance the forward transfer of RL training. Extensive evaluation shows that the proposed approach is able to well improve over PPO and an EWC based lifelong learning approach.

**Audience:**

Yes

**Broader Impact Concerns:**

I believe the broader impact statement in the original paper has addressed the possible concerns on the implications of this work.

**Claims And Evidence:**

Yes

**Requested Changes:**

In summary, I'm positive about the quality of this paper. Although the novelty is limited, the extensive evaluation results demonstrated the effectiveness of the proposed approach.

However, I suggest the authors clarify the training efficiency of the proposed approaches, and analyze when and where the approach could be a good choice. Also, a minor revision could be emphasizing why the EWC method is chosen rather than others.

**Strengths And Weaknesses:**

**Strengths**
1. This paper focuses on a very interesting and promising research topic, that is, lifelong learning for RL, which I think is important for solving a series of sequential decision making problems, especially when they have similar problem structures.
2. The proposed method is well motivated and technically sound. The description of background is fairly good, which can inspire experts or non-experts in the related domain.
3. The evaluation results are ample, which covers the comparison and a lot of ablation studies on discrete and continuous RL settings with various states and reward functions. The analyses are also sufficient to explain the results in a comprehensive way.
4. The shortcomings of the proposed approach are described carefully.

**Weaknesses**
1. The novelty of this paper is relatively low. The main contribution is the application of modulation based lifelong learning to RL, with the proposed schemes to mix the masking scores seem intuitive.
2. The baselines for most experiments are PPO and EWC$_{MH}$, which I think are weak. Additional state-of-the-art approaches could have been tested and compared.
3. The training efficiency is not clear. Since RL training for one task may cost a long time, it should be clarified how the proposed approach saves the training time, especially compared to the main baseline EWC$_{MH}$. Also, it should be analysed how the training cost would increase if the RL problem scales grows. These analyses at least let readers know when and where the approach is a better choice than other lifelong learning approaches.

---

> ### Author Response · Authors · 2023-04-13
> **Response to Reviewer JcNN**
>
> We thank the reviewer for the insightful and useful comments. Please see our response to the feedback below. All the requested changes have been addressed in the revised version of paper.
>
>
> **1. Low Algorithmic Novelty:** We revised the paper in the Introduction (“contributions” part) to better explain the novelty that we felt we did not adequately highlight before. To the best of our knowledge, this is the first study to (1) introduce learned modulating masks in LRL and show competitive advantage with respect to P&C, Online EWC, CLEAR, and IMPALA; (2) show the exploitation and the composition of previously learned knowledge in a lifelong reinforcement learning with learned modulating masks. In addition to a baseline masking approach with independent mask search, we introduced two new incremental mask composition approaches (LC and BLC) with different initialization of linear combination parameters. The analysis reveals different properties in the exploitation of previous knowledge while learning new tasks, either favoring knowledge re-use or exploration. While the mathematical formulation is not complex, and linear combinations of masks were used in previous studies, none of those previous studies used such a combination in LRL, nor attempted an increasing number of masks with the specific purpose of helping the search of new masks.
>
> **2. Limited Baselines:** To better highlight the use of baselines, and the motivations behind our experimental section, we rearranged the section to present first the comparison of our methods with well established baselines (P&C, Online EWC, CLEAR, and IMPALA): the ProcGen scenarios provides an assessment of the potential of the method in terms of performance, which is useful to determine that the approach is competitive and worthy of further investigation. We believe this is a fair selection of baselines, although not exhaustive. Following, rather than continue attempting to demonstrate absolute SoTA on all benchmarks and all baselines, we focus the analysis on the different behaviors of the suggested variations of the novel methods on a range of benchmarks (Minigrid, CT-graph, Continual World) with the aim to provide better understanding of the learning mechanisms and dynamics. For such purpose, EWC multi-head and PPO were selected to provide a reference with a lifelong learning and a non-lifelong learning approach: such different behaviors can be observed in Figures 3, 4, 5 and 6 with the left panels showing the difference between non-lifelong learning PPO and the rest, and the right panels providing a focus on lifelong learning approaches on the rapidity of learning.
>
>
> **3. Training Efficiency:** Thanks for bringing up this point. The training plot figures (the plot on the right side of Figures 2, 3, 4, 5 and 9) in the paper implicitly conveyed the training efficiency as we see the training curve average across all tasks and seeds. The mask methods have steeper curves on average indicating fewer training steps to learn tasks on average. In response to this comment, we added a further analysis of the time (in training steps) taken to achieve a certain level of performance for each task was conducted. The analysis has been added to the supplementary material in Appendix H (data reported in Tables 25 - 29) of the revised paper and referenced in Section 6.

---

### Review · Reviewer_YFzb · 2023-04-21

**Summary Of Contributions:**

This paper proposes to use modulating masks from the classification literature to improve performance in lifelong reinforcement learning. The paper contributes:
1) A specification to use modulating masks in the LRL, specifically using PPO and IMPALA, although the specification is sufficiently general to be used more broadly.
2) A variant which can exploit previous knowledge to learn new tasks.
3) And a set of empirical evaluations showing the improvements modulating masks can have over several competitors.

**Audience:**

Yes

**Claims And Evidence:**

No

**Requested Changes:**

I would like to see several modifications. Most critically,

1. The inclusion of how all hyperparameters were set.

2. An increase in the number of seeds used for many of the experiments. I would prefer less environments tested with a fuller set of empirical evaluations (i.e. more seeds, sweeps over hypers) rather than many environments all with comparisons with low statistical power.

3. Justification of the aggregation plots (aggregating over tasks) to artificially raise the number of samples in your confidence intervals.

4. A clear statement about using task boundaries, as done in your approach, and the limitations of such an assumption in the lifelong learning setting early in the paper. This was lightly touched on in section 7, but not quite enough in my opinion.




**Strengths And Weaknesses:**

# Strengths:
- The idea is well motivated and grounded in the continual learning domain in classification.
- The authors are very clear what the scope of the paper and the scope of contributions are. Specifically, the paper is very concrete in what hypotheses/methods it is testing.
- The algorithms developed in this paper are well explained, and easy to follow.

# Weaknesses:

1. It is common in continual learning to make an assumption that you have clear task boundaries. While this is a reasonable assumption for many applications, using the task information as privileged information (which your method does to the best of my understanding) needs to be justified and highlighted. Because this is a huge limitation of your approach and only covers a very small set of possible settings in LRL (as specified in [Khetarpal et al., 2020]. This comes up in the introduction, where this distinction isn't highlighted, and in the problem formulation section where the paper claims the lifelong setup is for a finite set of discrete tasks.

2. Statistical power of the hypothesis tests (i.e. the error bars) for each of these experiments. While 95% confidence intervals are used, the paper only has three seeds for all the experiments. In most of the plots we see heavy overlaps when there is no post-processing done to the results. When averaging over tasks (Fig 2 (right), Fig 3 (right), Fig 4 (right), Fig 5 (right), ...) we start to see an effect, but I'm not convinced this aggregation increases the statistical power of the hypothesis test. We are still only testing 3 sample per experiment, as the agent's random initialization is the primary source of randomness in the test. This is re-confirmed through the EWC's error bars (the main competitor) overlapping all other lifelong learning methods in Table 1, 2, etc... Because we are trying to do a performance comparison, I'm not very convinced by the results shown.

3. Given 2, and the huge difference between the variance in the modulating masks methods and EWC, I'm wondering how much of this is influenced by the hyperparameter tuning. It is unclear how hyperparameters are chosen for the experiments. This needs to be explained clearly, and choices need to be justified. This similarly seeds doubt in the continuos world experiments where EWC performs very poorly. Is this known behavior of EWC? Or is it specific to the hyperparameters chosen? This is a base question generally holding back the claims of the paper.

---

> ### Author Response · Authors · 2023-04-29
> **Response to Reviewer YFzb (1/2)**
>
> We thank the reviewer for the insightful and useful comments. Please see our response to the feedback below.
>
> Weaknesses
> **1. Clear Task Boundaries**
> We agree that the assumption of precise task boundaries and the use of task IDs is a limitation of all LRL methods that make this assumption. However, the nature of the proposed approach that associates a set of parameters (from a mask) to a task implies that task detection methods can be employed to address this limitation. The detection of task changes without an oracle can be implemented using probabilistic approaches [1], optimal transport methods [2] and other similar approaches being recently proposed. The addition of an automatic task labelling algorithm to the proposed method can extend its applicability in future studies. The limitations of task IDs and boundaries have been highlighted in Section 1, and further discussed in Section 7 of the revised paper.
>
> **2. Statistical power of the hypothesis tests**
> In these tests, the objective is to assess the learning performance of the algorithms when facing a new task. In the experiments of Figs. 3 to 6 (Right) of the revised paper, there are 8, 12, 10 and 10 distinct tasks respectively. Each task is learned three times with different seeds. This results in 24, 36, 30 and 30 distinct and different samples respectively. A non-aggregated version of the data in Fig.19 confirms that learning across tasks and seeds is distinct and different. We therefore strongly rebut the statement that we “artificially raise the number of samples”.
>
> Rather, an interesting point is that knowledge reuse in a LL curriculum means that task learning of later tasks may be affected by previous tasks, introducing a dependency among samples. However, this is precisely the intended objective of the $\mathrm{Mask_{LC}}$ and $\mathrm{Mask_{BLC}}$ variations that appear to perform well in those plots.
>
> To improve clarity when introducing the plots in Figs. 3 to 6(Right), we added the sentence “In these tests, the objective is to assess the learning performance of the algorithms when facing a new task. We do so by averaging all task performances across all seeds. A non-aggregated version of the data is shown in Fig. 19. Note that all algorithms except Mask_RI can exploit previous knowledge to learn new tasks.”
>
> **3. Hyper-parameters**
> The hyper-parameters for each method were set based on well-established values and preliminary tests. For the ProcGen curriculum, the hyper-parameters in [5] were followed. Across all other benchmarks, there were slight variations in the final set of hyper-parameters. Within each benchmark, the PPO hyperparameters were kept the same across all methods to ensure consistency and fair comparison. In line with the principles of providing meaningful baselines, for example, we observed that interfering tasks affected the performance of EWC when a single output head was employed. This led us to use instead a multi-head implementation of EWC. The preceding text about the hyperparameter selection, alongside the justification of the choice has been added to the supplementary material in Appendix B of the revised paper.
>
> The combination of PPO+EWC on lifelong continuous control problems has not been extensively studied. In a recent study [3], the combination also demonstrated sub-optimal performance in another lifelong continuous control benchmark. We speculate the regularization of the network parameters by EWC could impede on exploration, a known challenge in continuous control problems [4]. To maintain consistency and make the approaches comparable, we chose to maintain the same baselines across the CT-graph, Minigrid and Continual World environments. We do not believe that the poor performance of PPO+EWC method in the continuous control environment is holding back the claims of the paper: on the contrary, it supports the flexibility of parameter-isolation based methods as a flexible approach in combination with PPO for both discrete and continuous control environments.

---

> ### Author Response · Authors · 2023-04-29
> **Response to Reviewer YFzb (2/2)**
>
> Requested Changes
> **1. Hyper-parameter settings**
> Please see response 3 in the weaknesses section.
>
> **2. Increase number of seeds**
> We agree that more seeds are always desirable. We are currently evaluating how many more runs we can perform given the computational resources and time available. We believe we can run 2 extra seeds for the CT-graph curricula, which involve a total of 36 additional experiments (2 seeds times 3 curricula times 5 methods + 1 STE). Each takes between one and two hours each, which should enable us to complete the runs and update the paper by the end of the rebuttal period.
>
> We would also like to point out that while this reviewer stresses the importance of multiple seeds, other reviewers have pointed out the utility of a range of different benchmarks, which contributes to generally assess the versatility of the method in different domains (e.g., both discrete and continuous control environments). We assure the reviewer that we are trying our best to provide as much significance as possible, and also maintain a focus on the investigation of the properties of the proposed method, rather than on proving absolute best performance with respect to other baselines.
>
> **3. Justification of aggregated plots**
> Please see response 2 in the weaknesses section.
>
> **4. Task boundaries**
> Please see response 1 in the weaknesses section.
>
>
> **References**
> [1] Milan, K., Veness, J., Kirkpatrick, J., Bowling, M., Koop, A. and Hassabis, D., 2016. The forget-me-not process. Advances in Neural Information Processing Systems, 29.
>
> [2] Liu, X., Bai, Y., Lu, Y., Soltoggio, A. and Kolouri, S., 2022. Wasserstein Task Embedding for Measuring Task Similarities. arXiv preprint arXiv:2208.11726.
>
> [3] Mendez JA, van Seijen H, Eaton E. Modular lifelong reinforcement learning via neural composition. arXiv preprint arXiv:2207.00429. 2022 Jul 1.
>
> [4] Lillicrap TP, Hunt JJ, Pritzel A, Heess N, Erez T, Tassa Y, Silver D, Wierstra D. Continuous control with deep reinforcement learning. arXiv preprint arXiv:1509.02971. 2015 Sep 9.
>
> [5] Powers S, Xing E, Kolve E, Mottaghi R, Gupta A. Cora: Benchmarks, baselines, and metrics as a platform for continual reinforcement learning agents. In Conference on Lifelong Learning Agents 2022 Nov 28 (pp. 705-743). PMLR.

---

### Author Response · Authors · 2023-05-05
**Summary of our Revision**

Dear all,

We thank the reviewers for the insightful comments and feedback. We addressed the comments and complied to the requested changes to the best of our ability. Comments on novelty, baselines, task boundaries, training efficiency and more have been discussed in the responses and used to edit the paper accordingly. The updates are available in the revised paper that, as a consequence, has improved according to the reviewers’ recommendations.

Thanks,
Authors

---

### Decision · Action_Editors · 2023-06-19

**Recommendation:** Accept as is

**Comment:**

(See meta-review above).

**Audience:**

(See meta-review above).

**Claims And Evidence:**

**Meta Review for Lifelong Reinforcement Learning with Modulating Masks**

As summarized by reviewer JcNN and others: This research paper introduces a novel approach to lifelong learning in the realm of reinforcement learning (RL) using modulation techniques. It presents several strategies for learning masking scores for each task, including Mask Random Initialization, Mask Linear Combination, and a hybrid approach called Balanced Linear Combination. These schemes aim to effectively utilize previously trained masking scores to enhance the transfer of RL training in a forward manner. Through extensive evaluation, the proposed approach demonstrates significant improvements over PPO and an EWC-based lifelong learning approach.

Key Strengths: This paper addresses a fascinating and promising research area, namely lifelong learning in RL, which is crucial for tackling a range of sequential decision-making problems, particularly when they share similar structural characteristics. The proposed method is well-motivated and technically sound. The background information provided is comprehensive and can inspire both experts and non-experts in the field. The evaluation results are extensive, encompassing comparisons and numerous ablation studies conducted on discrete and continuous RL settings with diverse state and reward functions. The analysis provided adequately explains the results in a comprehensive manner. Furthermore, the limitations of the proposed approach are carefully described.

Reviewers have raised concerns regarding (1) limited novelty of the work and the limitations of the method, (2) evaluation issues and baselines, (3) experimental setup (4) clarity, presentation and communication of the results.

The authors have devoted the effort to address each point reviewers made, and have incorporated some of the feedback into improving the manuscript to address the weaknesses. Two of Three reviewers were satisfied with the author’s rebuttal and improvements, and after carefully studying the points raised by reviewer Cewb and the responses by the authors, I’m siding on accepting the work, as I believe it has something of value to offer to the RL community, the audience of the work.